# HARNESSING ATTENTION PRIOR FOR REFERENCE-BASED MULTI-VIEW IMAGE SYNTHESIS

## ABSTRACT

This paper explores the domain of multi-view image synthesis, aiming to create specific image elements or entire scenes while ensuring visual consistency with reference images. We categorize this task into two approaches: local synthesis, guided by structural cues from reference images (Reference-based inpainting, Ref-inpainting), and global synthesis, which generates entirely new images based solely on reference examples (Novel View Synthesis, NVS). In recent years, Text-to-Image (T2I) generative models have gained attention in various domains. However, adapting them for multi-view synthesis is challenging due to the intricate correlations between reference and target images. To address these challenges efficiently, we introduce Attention Reactivated Contextual Inpainting (ARCI), a unified approach that reformulates both local and global reference-based multi-view synthesis as contextual inpainting, which is enhanced with pre-existing attention mechanisms in T2I models. Formally, self-attention is leveraged to learn feature correlations across different reference views, while cross-attention is utilized to control the generation through prompt tuning. Our contributions of ARCI, built upon the StableDiffusion fine-tuned for text-guided inpainting, include skillfully handling difficult multi-view synthesis tasks with off-the-shelf T2I models, introducing task and view-specific prompt tuning for generative control, achieving end-to-end Ref-inpainting, and implementing block causal masking for autoregressive NVS. We also show the versatility of ARCI by extending it to multi-view generation for superior consistency with the same architecture, which has also been validated through extensive experiments.

## 1 INTRODUCTION

This paper delves into the intricate domain of multi-view image synthesis, with a central focus on crafting specific image elements or complete scenes by leveraging reference images as the foundation. The objective is to generate specific components or even entire images through these references, while meticulously preserving visual coherence in aspects such as geometry, color, and texture between the reference images and their synthesized counterparts. This task can be broadly categorized into two facets: local and global multi-view image synthesis from reference images. The local variant involves the creation of specific image segments by aligning with the locally inherent structural cues found in the reference images. This technique is essentially related to a previously defined concept known as Reference-guided inpainting (Ref-inpainting) (Zhou et al., 2021; Zhao et al., 2022b), as illustrated in Fig. 1(a). Conversely, the global multi-view image synthesis aims to generate entirely new images, drawing inspiration solely from reference examples, as depicted in Fig. 1(b). This approach is closely associated with Novel View Synthesis (NVS) (Liu et al., 2023b). In this paper, we introduce a unified methodology to tackle this task by reactivating self-attention priors derived from extensive text-to-image models (Rombach et al., 2022), as illustrated in Fig. 2(a).

In recent years, Text-to-Image (T2I) generative models have garnered substantial attention across various domains, finding applications in diverse areas such as personalization (Gal et al., 2022; Ruiz et al., 2022; Mokady et al., 2022), controllable image-to-image generation (Zhang & Agrawala, 2023; Mou et al., 2023; Yang et al., 2023; Bar-Tal et al., 2023), and even 3D generation (Poole et al., 2023; Lin et al., 2023; Liu et al., 2023b). While it may seem intuitive to harness the power of T2I generative models to directly address multi-view synthesis by training additional adapters with zero-initialization, as demonstrated in previous works (Hu et al., 2021; Zhang & Agrawala, 2023; Mou

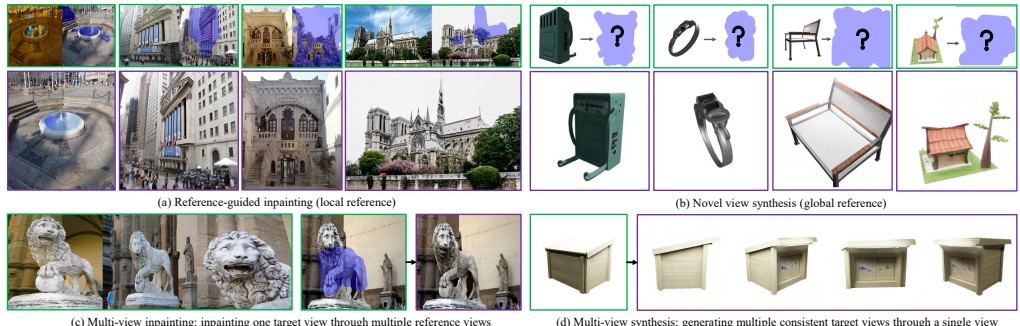

(a) Reference-guided inpainting (local reference)

(b) Novel view synthesis (global reference)

(c) Multi-view inpainting: inpainting one target view through multiple reference views

(d) Multi-view synthesis: generating multiple consistent target views through a single view

Figure 1: Illustration of local and global reference-based multi-view synthesis addressed by ARCI.

et al., 2023), these adapter-based fine-tuning strategies have inherent limitations. They struggle to capture fine-grained correlations, including object orientations and precise object locations, between reference and target images. These nuanced details are pivotal for tasks such as multi-view generation, as exemplified by Ref-inpainting. Furthermore, leveraging T2I models to address image-to-image tasks requires image-based guidance rather than text-based ones. Thus some approaches replace textual encoders with visual ones and optimize them for full fine-tuning of the entire T2I model. This transition often demands substantial computational resources, with training sessions extending to hundreds of GPU hours, as in Yang et al. (2023); Liu et al. (2023b). However, training these large T2I models with unfamiliar visual encoders can be computationally intensive and challenging to converge, particularly when working with limited batch sizes. Additionally, most visual encoders, such as image CLIP (Radford et al., 2021), tend to emphasize the learning of semantic features rather than the intricate spatial details essential for tasks involving multi-view synthesis.

Our innovative approach is designed to harness off-the-shelf T2I generative models to tackle both local and global multi-view synthesis, overcoming the aforementioned challenges of extensive training cost and imperfect image encoding. It stems from a profound realization: most T2I models inherently incorporate attention mechanisms adept at discerning spatial correlations within images and text. These self and cross-attention, originally acquired through training with large diffusion generative models, can serve as an intrinsic guiding prior to multi-view image synthesis.

This leads us to a pivotal inquiry: "*Could pre-existing attention mechanisms have already established meaningful correlations between reference images and the intended generative targets?*" To leverage this untapped potential, we introduce Attention Reactivated Contextual Inpainting (ARCI), a unified approach that encompasses both reference-based local and global multi-view synthesis. ARCI ingeniously reformulates reference-based multi-view synthesis as a contextual inpainting process. This involves seamlessly integrating the reference conditions and masked targets into a unified tensor within the self-attention module as in Fig. 2(a). We then employ pre-trained textual encoders and cross-attention modules to guide the generation of T2I models, infusing them with critical information for specific generative tasks and desired view orders. The contextual inpainting (Yu et al., 2018) was originally proposed to leverage attention to infill missing features for inpainting, sharing some similarities to our work, which inpaints target views through information aggregated from references.

Formally, ARCI represents an innovative approach, built upon the StableDiffusion (SD) (Rombach et al., 2022)[1] fine-tuned under text-guided inpainting. ARCI is primarily designed to address a diverse range of image synthesis tasks, including Ref-inpainting and NVS from reference images. Importantly, these tasks can be seamlessly extended into the multi-view scenario, as depicted in Fig. 1(c)(d). Additionally, we introduce task and view-specific prompt tuning to effectively control generative tasks and define specific view orders. Remarkably, even for the prohibitive Ref-inpainting task, which typically demands sophisticated 3D geometrical warping and 2D inpainting techniques (Zhou et al., 2021; Zhao et al., 2022b;a), our ARCI framework addresses it end-to-end with minimal additional parameters while keeping all other weights from SD frozen. On the other hand, to tackle the more intricate NVS task, we propose the novel technique of block causal masking, facilitating self-attention-based T2I models in achieving consistent autoregressive generation as in Fig. 1(d).

The novel contextual formulation of ARCI results in faster convergence and efficient training, outperforming other baselines (Zhang & Agrawala, 2023; Yang et al., 2023; Liu et al., 2023b) with

---

[1]StableDiffusion 2.0: https://github.com/Stability-AI/stablediffusion.

the same training computation and parameters. Moreover, the introduction of task and view-specific prompt tuning marks a significant advancement in the field, allowing us to efficiently control the generation of T2I models for multi-view synthesis.

We highlight the key contributions as follows: 1) *Efficient multi-view synthesis with T2I models:* Benefiting from the novel contextual inpainting formulation and inherent attention mechanisms from generative T2I models, the ARCI provides an efficient solution for multi-view synthesis without thoroughly laborious re-training T2I models. 2) *Task and view-specific prompt tuning:* Our work pioneers the use of task and view-specific prompt tuning, allowing for precise control over generative tasks and view orders. 3) *End-to-end Ref-inpainting:* Notably, our ARCI addresses the challenging Ref-inpainting end-to-end, without complex 3D geometrical warping and 2D inpainting techniques. 4) *Autoregressive NVS with block causal masking*: For the intricate NVS task, we introduce the novel concept of block causal masking, enabling self-attention-based T2I models to achieve Autoregressive generation for superior quality and geometric consistency.

## 2 RELATED WORK

**Personalization and Controllability of T2I Models.** Recent achievements on T2I have produced impressive visual generations, which could be further extended into local editing (Avrahami et al., 2022; Hertz et al., 2022; Couairon et al., 2022). However, these T2I models could only be controlled by natural languages. As "an image is worth hundreds of words", T2I models based on natural texts fail to produce images with specific textures, locations, identities, and appearances (Gal et al., 2022). Textual inversion (Gal et al., 2022; Mokady et al., 2022) and fine-tuning techniques (Ruiz et al., 2022) are proposed for personalized T2I. Meanwhile, many works pay attention to image-guided generations (Voynov et al., 2022; Li et al., 2023; Ma et al., 2023). ControlNet (Zhang & Agrawala, 2023) and T2I-Adapter (Mou et al., 2023) learn trainable adapters (Houlsby et al., 2019) to inject visual clues to pre-trained T2I models without losing generalization and diversity. But these moderate methods only work for simple style transfers. More spatially complex tasks, such as Ref-inpainting, are difficult to handle by ControlNet as verified in Sec. 4. In contrast, T2I-based exemplar-editing and NVS have to be fine-tuned on large-scale datasets with strong data augmentation (Yang et al., 2023) and large batch size (Liu et al., 2023b). Compared with these aforementioned manners, the proposed ARCI enjoys both spatial modeling capability and computational efficiency.

**Prompt Tuning** (Lester et al., 2021; Liu et al., 2021b;a) indicates fine-tuning token embeddings for transformers with frozen backbone to preserve the capacity. Prompt tuning is first explored for adaptively learning suitable prompt features for language models rather than manually selecting them for different downstream tasks (Liu et al., 2023a). Moreover, prompt tuning has been further investigated in vision-language models (Radford et al., 2021; Ge et al., 2022) and discriminative vision models (Jia et al., 2022; Liao et al., 2023). Visual prompt tuning in Sohn et al. (2022) prepends trainable tokens before the visual sequence for transferred generations. Though both ARCI and Sohn et al. (2022) aim to tackle image synthesis, our prompt tuning is used for controlling text encoders rather than visual ones. Thus ARCI enjoys more intuitive prompt initialization from task-related textual descriptions. Besides, we should clarify that prompt tuning is different from textual inversion (Gal et al., 2022). In our work, we use prompt tuning to address specific downstream tasks with view information, while textual inversion tends to present personalized image subsets.

**Reference-guided Image Inpainting.** Image inpainting is a long-standing vision task, which aims to fill missing image regions with coherent results. Both traditional methods (Bertalmio et al., 2000; Criminisi et al., 2003; Hays & Efros, 2007) and learning-based ones (Zeng et al., 2020; Zhao et al., 2021; Li et al., 2022; Suvorov et al., 2022; Dong et al., 2022) achieved great progress in image inpainting. Furthermore, Ref-inpainting requires recovering a target image with one or several reference views (Oh et al., 2019), which is useful for repairing old buildings or removing occlusions in popular attractions. But Ref-inpainting usually suffers from a sophisticated pipeline (Zhou et al., 2021; Zhao et al., 2022b;a), including depth estimation, pose estimation, homography warping, and single-view inpainting. Note that for large holes, the geometric pose is not reliable; and the pipeline will be largely degraded. Thus an end-to-end Ref-inpainting pipeline is meaningful. To the best of our knowledge, we are the first ones to tackle such a difficult reference-guided task with T2I models.

**Novel View Synthesis from a Single Image.** NVS based on a single image is an intractable ill-posed problem, requiring both sufficient geometry understanding and expressive textural presentation (Fahim

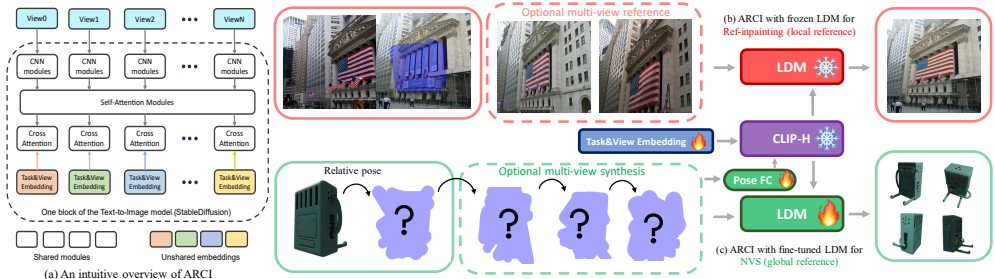

Figure 2: (a) Overview: ARCI uses a shared T2I model to learn correlations across multi-view images with concise designs and fast convergence. Detailed architecture for (b) local reference-based Ref-inpainting and (c) global reference-based NVS. ARCI also supports optional multi-view Ref-Inpainting and consistent NVS.

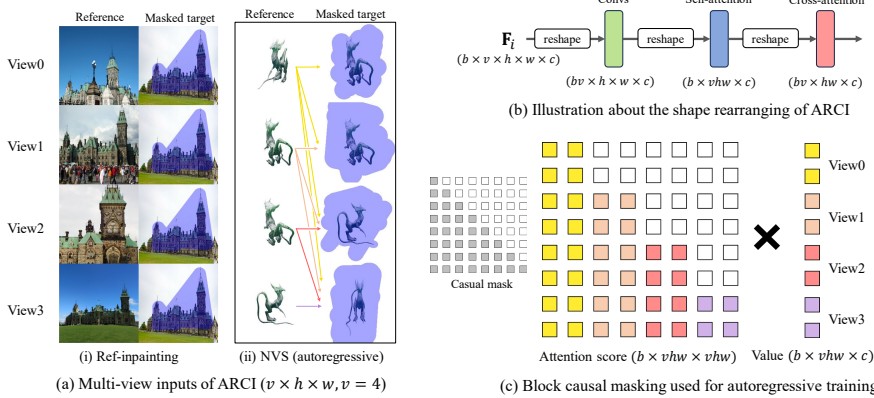

Figure 3: Illustration about the (a) multi-view training inputs, (b) feature rearranging, and (c) block causal masking. All views of Ref-inpainting (a-i) share the same masked target, while the multi-view NVS (a-ii) should be trained with the AR generation. $v, h, w$ indicate the view number, height, and width of images or features.

et al., 2021). Many previous works could partially tackle this problem through single view 3D reconstruction (Wu et al., 2017; Wang et al., 2018; Chen et al., 2019; Liu et al., 2019; Xu et al., 2019), 2D generative models (Niklaus et al., 2019; Shih et al., 2020; Rombach et al., 2021), feature cost volumes (Chan et al., 2023), and GAN-based methods (Schwarz et al., 2020; Niemeyer & Geiger, 2021; Chan et al., 2022). However, these manners still suffer from limited generalization or small angle variations. Recent works devoted to incorporating the knowledge from 2D diffusion-based T2I models into the 3D reconstruction (Poole et al., 2023; Wang et al., 2023; Lin et al., 2023; Tang et al., 2023; Qian et al., 2023), largely alleviating the demand for 3D annotated data. On the other hand, Zero123 (Liu et al., 2023b) uses another image CLIP encoder to inject image features from the reference view for NVS. Although Zero123 unlocks the view synthesis capability of T2I models, it requires a large batch size and expensive computational resources to stabilize the training stage with an unknown reference image encoder. Moreover, the image encoder in Zero123 can only tackle one reference image, which fails to generate consistent multi-view images.

## 3 ATTENTION REACTIVATED CONTEXTUAL INPAINTING

### 3.1 FRAMEWORK OF ARCI

**Overview.** Our ARCI model is formulated in Sec.3.1. Then we explain using attention prior to learn multi-view correspondence Sec.3.2, as well as enhancing self-attention for AR generation. Finally, we discuss the task and view-specific prompt tuning for cross-attention modules in Sec. 3.3.

As shown in Fig. 2, our ARCI framework is built upon the inpainting pre-trained LDM (Rombach et al., 2022). Essentially, we reformulate both Ref-inpainting and NVS as contextual image inpainting problems. The conventional approach to integrating reference images into the generation process involves either concatenating them along the channel dimension of input images or introducing an additional image encoder for the network (Yang et al., 2023; Zhang & Agrawala, 2023; Mou et al., 2023; Liu et al., 2023b). However, both strategies require additional training to rebuild the correlation

between the reference and target for the generative model. This is contrary to our motivation to harness the potential of large T2I models with minimal architectural alterations.

Thanks to the convolutional U-net architecture in LDM, we can expand the input image in the spatial dimensions without extra modification. To illustrate, let's consider a scenario with a single reference image at first. Our input $\mathbf{I}'$ is a stitching image of $\mathbf{I}_{ref}$ and the masked target $\hat{\mathbf{I}}_{tar}$, forming as $\mathbf{I}' = [\mathbf{I}_{ref}; \hat{\mathbf{I}}_{tar}] \in \mathbb{R}^{H \times 2W}$. In practice, we take the reference image on the left side, while the target one is placed on the right side. For the multi-view references, we stitch each reference with the specific target as shown in Fig. 3(a). All views are learned separately for convolutions and cross-attention, while they share the same self-attention processing as discussed in Sec. 3.2. As shown in Fig. 3(a-i), the multi-view Ref-inpainting leverages information from different reference views to repair the same target one, while multi-view NVS could be seen as an AR generation for sequentially consistent view synthesis. Masked targets could be generated successfully along with the ARCI filling the hole in the target side. The inpainting formulation fully exploits the contextual learning capability of pre-trained self-attention in LDM to address the reference-guided generation. More details about the processing of data and masks are discussed in Appendix. A.2.

Simply stitching reference images and masked targets along the spatial space and self-attention modules could not establish the necessary correlation between them, thus failing to generate the desired target results. Text-guided inpainting needs image-dependent text prompts to drive the diffusion model for the desired generation. It is non-trivial to define Ref-inpainting and NVS with natural languages. On the other hand, it is beneficial to have an image-independent prompt to guide the diffusion model to perform specific tasks. To this end, we propose to use prompt tuning to learn task and view-specific prompts as in Sec. 3.3. Except for the prompt tuning, all weights in LDM are frozen in Ref-inpainting to maintain the proper generalization as shown in Fig. 2(b). For the NVS, we need to fine-tune the whole LDM as in Fig. 2(c), but we clarify that ARCI enjoys much better convergence compared with other fine-tuning based methods, such as Zero123 (Liu et al., 2023b).

## 3.2 REACTIVATING SELF-ATTENTION MODULES

As shown in Fig. 3(b), given multi-view features $\mathbf{F}_i \in \mathbb{R}^{b \times v \times h \times w \times c}$ to layer $i$ with $v$ views, $b, h, w, c$ indicate the batch size, feature height, width, and channels respectively, all MLP, convolutional, and cross-attention layers handle $\mathbf{F}_i$ separately. We could easily achieve this by reshaping the view and batch dimension together as $\hat{\mathbf{F}}_i \in \mathbb{R}^{bv \times h \times w \times c}$ or $\hat{\mathbf{F}}_i \in \mathbb{R}^{bv \times hw \times c}$. Before the self-attention encoding, we adjust the feature shape as $\tilde{\mathbf{F}}_i \in \mathbb{R}^{b \times vhw \times c}$, thus features across different views could be learned together. To further incorporate positional clues to T2I models for distinguishing different sides of reference and target, we incrementally add positional encoding $P_i$ to $\mathbf{F}_i$ as

$$P_i = \gamma_i \cdot \text{cat}([P_v; P_{Fourier}]), \tag{1}$$

where $P_v, P_{Fourier}$ indicate the trainable view embedding and Fourier absolute positional encoding (Vaswani et al., 2017) respectively; $\gamma_i$ is a zero-initialized learnable coefficient for each layer.

For the Ref-inpainting, no masking strategy should be considered in self-attention modules. All reference views share the same target one, thus it is unnecessary for ARCI to sequentially repair target views. In contrast, generating consistent novel views from a single image needs our model to handle the sequential generation with dynamic reference views. For example, the same ARCI should accomplish the NVS from one view, two views, and even more. So the AR generation (Van den Oord et al., 2016; Salimans et al., 2017; Esser et al., 2021) is suitable to formulate this task.

**Block Causal Masking.** Although fine-tuning ARCI mitigates heavy training costs of T2I models, it still needs a certain fine-tuning for LDM to effectively tackle challenging NVS as shown in Fig. 2(c). The intuitive solution is to train an AR-based generative model that can generalize across various view numbers for multi-view synthesis. Converting a pre-trained diffusion model to an AR-based generative model is non-trivial. However, the inpainting formulation utilized by ARCI makes this conversion feasible. Specifically, we just need to adjust the masking strategy during the self-attention learning. We propose the block casual masking as shown in Fig. 3(c), while the block size of each view is $h \times w$, matching the size of the stitched reference and target pair. Different from the traditional casual mask which is a lower triangular matrix, the block casual mask enlarges the minimal unit from one token to a $h \times w$ block, ensuring reasonable block-wise receptive fields. In practice, all uncolored

Table 1: Quantitative results for Ref-inpainting on MegaDepth (Li & Snavely, 2018) test set based on matching and manual masks with 1-view reference (upper). Lower results are based on the multi-view reference-based test set (Appendix. A.2.1). 'ExParams' means the scale of extra trainable parameters. * means that the uncorrupted ground truth is visible for the matching.

| Methods | PSNR↑ | SSIM↑ | FID↓ | LPIPS↓ | ExParams |
|---|---|---|---|---|---|
| SD (inpainting) (Rombach et al., 2022) | 19.841 | 0.819 | 30.260 | 0.1349 | +0 |
| ControlNet (Zhang & Agrawala, 2023) | 19.072 | 0.744 | 33.664 | 0.1816 | +364.24M |
| ControlNet+New Cross-Attention | 19.027 | 0.743 | 34.170 | 0.1805 | +463.41M |
| ControlNet+Matching* (Tang et al., 2022) | 20.592 | 0.763 | 29.556 | 0.1565 | +364.29M |
| Perceiver+ImageCLIP (Jaegle et al., 2021) | 19.338 | 0.745 | 32.911 | 0.1751 | +52.01M |
| Paint-by-Example (Yang et al., 2023) | 18.351 | 0.797 | 34.711 | 0.1604 | +865.90M |
| TransFill (Zhou et al., 2021) | **22.744** | **0.875** | 26.291 | 0.1102 | – |
| ARCI | 20.926 | 0.836 | **18.680** | **0.0961** | +0.05M |
| ARCI (1-view) | 21.195 | 0.837 | 18.598 | 0.0946 | +0.05M |
| ARCI (2-view) | 21.092 | 0.836 | 18.389 | 0.0969 | +0.055M |
| ARCI (3-view) | 21.356 | 0.840 | 16.838 | 0.0901 | +0.06M |
| ARCI (4-view) | **21.779** | **0.847** | **16.632** | **0.0839** | +0.065M |

tokens in the attention score are masked with "$-\inf$" before the softmax operation. We should clarify the block casual mask can be achieved parallelly and efficiently as discussed in Alg. 1 of Appendix.

### 3.3 TASK&VIEW PROMPT TUNING TO REACTIVATE CROSS-ATTENTION

The prompt embedding is adopted as the textual branch of Fig. 2. Specifically, we prepare a set of trainable text embeddings for different generative tasks, which are further categorized into task and view prompts. Specifically, task prompt embeddings are shared in the same task, *e.g.*, all views of Ref-inpainting using the same task embeddings. In contrast, view prompt embeddings are applied to different views to inject different view information through cross-attention modules. Though there are only a few trainable parameters (0.05M to 0.065M), we astonishingly find that prompt tuning is sufficient to drive complex generative tasks such as Ref-inpainting, even though the LDM backbone is completely frozen. The trainable task and view prompt embeddings $p_t, p_v$ are initialized as the averaged embedding of the natural task description. The optimization target can be defined as

$$\{p_t, p_v\}_* = \underset{\{p_t, p_v\}}{\arg\min} \, \mathbb{E}_{x, \epsilon \sim \mathcal{N}(0, \mathcal{I}), t} \left[ \| \varepsilon - \varepsilon_\theta \left( [z_t; \hat{z}_0; \mathbf{M}], c_\phi(p_t, p_v), t \right) \|^2 \right], \tag{2}$$

where $\varepsilon_\theta(\cdot)$ is the estimated noise by LDM; $c_\phi(\cdot)$ means the frozen CLIP-H; $z_t$ is a noisy latent feature of input $z_0$ in step $t$; $\hat{z}_0 = z_0 \odot (1 - \mathbf{M})$ are masked latent features that are concatenated to $z_t$ with mask $\mathbf{M}$. Task and view-specific prompt tuning enjoy not only training efficiency but also lightweight saving (Lester et al., 2021). For example, we could address Ref-inpainting for different reference views with the same ARCI, while only 0.01% additional weights of $\{p_t, p_v\}_*$ are required to be changed for each view condition. In NVS, we further provide relative poses to ARCI. Following Liu et al. (2023b), we calculate the 4-channel relative pose in the polar coordinate from the first view to the target one for each view, which is encoded by a two-layer FC. Then the pose feature replaces the last padding token in the prompt embeddings before being applied to the CLIP-H.

## 4 EXPERIMENTS

**Datasets.** For Ref-inpainting, we use the resized 512×512 image pairs from MegaDepth (Li & Snavely, 2018), which includes many multi-view famous scenes collected from the Internet. To trade-off between the image correlation and the inpainting difficulty, we empirically retain image pairs with 40% to 70% co-occurrence with about 80k images and 820k pairs. The validation of Ref-inpainting also includes some manual masks from ETH3D scenes (Schops et al., 2017) to verify the generalization. For the NVS, we use Objaverse (Deitke et al., 2022) rendered by Liu et al. (2023b) including 800k various scenes with object masks. We resize all images to 256×256 as Liu et al. (2023b). Note that some extreme views with elevation angles less than -10° are filtered due to excessive ambiguity. More details about the masking and datasets are in Appendix. A.2.

**Implementation Details.** Our ARCI is based on the inpainting fine-tuned SD (Rombach et al., 2022) with 0.8 billion parameters. For the task and view prompt tuning, there are 50 trainable prompt tokens

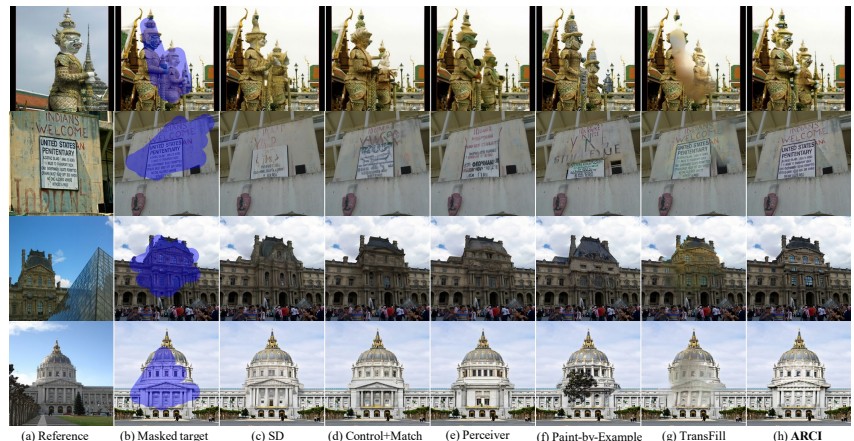

(a) Reference  (b) Masked target  (c) SD  (d) Control+Match  (e) Perceiver  (f) Paint-by-Example  (g) TransFill  (h) **ARCI**

Figure 4: Qualitative Ref-inpainting results on MegaDepth. More results are in Fig. 14 of Appendix.

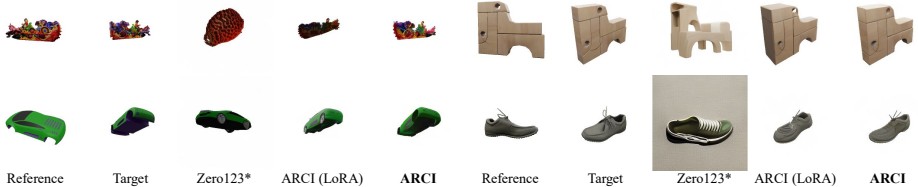

Reference  Target  Zero123*  ARCI (LoRA)  **ARCI**  Reference  Target  Zero123*  ARCI (LoRA)  **ARCI**

Figure 5: NVS results on Objaverse (Deitke et al., 2022) from a single reference image. Note that Zero123* was re-trained with a small batch size as our ARCI. More results are in Fig. 17 of Appendix.

at all. We use 90% tokens to represent the task embeddings, while 10% tokens indicate each view respectively. We use the AdamW optimizer with a weight decay of 0.01 to optimize ARCI. For the Ref-inpainting, the prompt tuning's learning rate is 3e-5. Moreover, 75% masks are randomly generated, and 25% of them are matching-based masks (Appendix. A.2.1). For the NVS, ARCI could be tested with the adaptive masking strengthened by the foreground segmentation model as in Appendix. A.3.1. ARCI could be converged with just 48 batch size and learning rate 1e-5. The multi-view ARCI is fine-tuned based on the one-view version. We also train a multi-view ARCI with 512 batch size and learning rate 3e-5 for NVS. More training details are listed in the Appendix. A.2.3.

## 4.1 RESULTS OF REFERENCE-GUIDED INPAINTING

**Results of One-view Reference.** We thoroughly compared the specific Ref-inpainting method (Zhou et al., 2021) and existing image reference-based variants of LDM with one-view reference in Tab. 1 and Fig. 4. Note that ControlNet (Zhang & Agrawala, 2023) fails to learn the correct spatial correlation between reference images and masked targets, even enhanced with trainable cross-attention learned between reference and target features. Furthermore, we try to warp ground-truth latent features with image matching (Tang et al., 2022) as the reference guidance for ControlNet, but the improvement is not prominent. Perceiver (Jaegle et al., 2021) and Paint-by-Example (Yang et al., 2023) align and learn image features from Image CLIP. Since image features from CLIP contain high-level semantics, they fail to deal with the fine-grained Ref-inpainting as shown in Fig. 4(e)(f). Though TransFill (Zhou et al., 2021) achieves proper results in PSNR and SSIM, it suffers from blur and color difference as in Fig. 4(g) with challenging viewpoints. ARCI enjoys substantial advantages in both qualitative and quantitative comparisons with negligible trainable weights. We further compare ARCI with TransFill on the officially provided real-world dataset in Tab. 9 and Fig. 13 of the Appendix. Since most instances should be defined as object removal tasks without ground truth, quantitative metrics are for reference only. But ARCI still outperforms TransFill in FID and LPIPS with perceptually pleasant results. Moreover, as shown in Fig. 13, ARCI enjoys good generalization in unseen or occluded regions, because it gets rid of the constrained geometric warping.

**Results of Multi-view Ref-inpainting.** We verified models trained with different numbers of reference views in Tab. 1. The 2-view ARCI suffers from a minor performance decrease, which we attribute to the inherent ambiguities within the MegaDepth validation. However, as the number

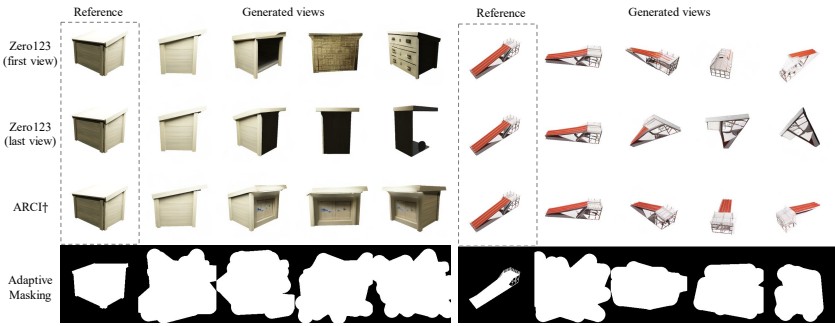

Figure 6: The sequential generative results from a single view. Results of Zero123 (Liu et al., 2023b) are conditioned on the real reference (first view) and the last generated view (last view) respectively.

Table 2: Results for NVS with a single image on Objaverse (Deitke et al., 2022). Zero123 (Liu et al., 2023b) was tested with the official model, while Zero123* was re-trained with the same training setting as ARCI. ARCI† is trained with a larger batch size (512).

| Methods | Ref-View | PSNR↑ | SSIM↑ | LPIPS↓ | CLIP↑ |
|---|---|---|---|---|---|
| Zero123* (re-trained) | 1 | 14.316 | 0.802 | 0.3455 | 0.6549 |
| Zero123 (Liu et al., 2023b) | 1 | 19.402 | 0.858 | 0.1309 | **0.7816** |
| ARCI (prompt tuning) | 1 | 16.385 | 0.855 | 0.2468 | 0.7107 |
| ARCI (LoRA) | 1 | 19.514 | 0.869 | 0.1534 | 0.7589 |
| ARCI | 1 | **20.508** | **0.875** | **0.1288** | 0.7763 |
| ARCI† | 1 | 21.551 | 0.885 | 0.1115 | 0.7927 |
| ARCI† | 2 | 22.680 | 0.893 | 0.0888 | 0.8247 |
| ARCI† | 3 | 23.839 | 0.905 | 0.0740 | 0.8372 |
| ARCI† | 4 | **24.064** | **0.906** | **0.0671** | **0.8468** |

of reference views increases, there is a notable enhancement in inpainting capability. As shown in Fig. 15, more consistent references lead to robust inpainting results with sensible structures.

## 4.2 RESULTS OF NOVEL VIEW SYNTHESIS

**Results of single-view NVS.** Different from Zero123 (Liu et al., 2023b), we explore the feasibility of learning the challenging NVS with limited resources as in the upper of Tab. 2 and Fig. 5. The CLIP score (Radford et al., 2021) is compared to evaluate the similarity between the generation and the target. Prompt tuning and LoRA-based (Hu et al., 2021) fine-tuning are insufficient to achieve high-fidelity results as in Tab. 2. Fine-tuning the whole LDM enjoys substantial improvements. But all variants of ARCI outperform the state-of-the-art competitor Zero123 with the same training setting (batch size 48 with 2 A6000 GPUs) in Tab. 2 because the latter needs a much larger batch size (1536) for stable training. Thus our contextual inpainting-based ARCI enjoys a good balance between training efficiency and performance. Moreover, we evaluate the effectiveness of Classifier-Free Guidance (CFG) (Ho & Salimans, 2022) during the training phase, which enjoys better pose guidance with a CFG coefficient of 2.5 for the inference. We further provide the training log of ARCI and Zero123 in Fig. 9 of the Appendix. Obviously, the contextual inpainting-based ARCI enjoys a substantially faster convergence and superior performance.

**Results of Multi-view NVS.** We further fine-tune the ARCI for multi-view NVS through the AR generation and large batch size (ARCI†). Quantitative results are shown in the lower of Tab. 2. Obviously, more reference views lead to better reconstruction quality of ARCI†. Moreover, additional reference images could substantially alleviate the ambiguity, improving the final results with consistent geometry as shown in Fig. 18. Benefited by the AR, ARCI† can be also generalized to synthesize a group of consistent images with different viewpoints from a single view as shown in Fig. 6 and Fig. 10. We also show that our method can be generalized to real-world data as in Fig. 11.

## 4.3 ANALYSIS AND ABLATION STUDIES

**Self-Attention Analysis.** We show the visualization of self-attention scores attended by masked regions for Ref-inpainting (Alg. 2) across different DDIM steps in Fig. 7. Self-attention can success-

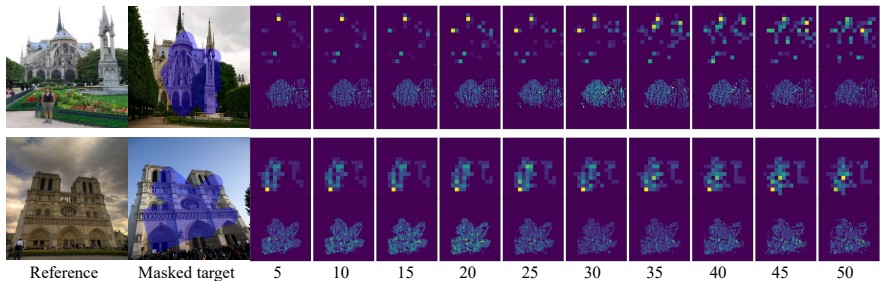

Reference   Masked target   5   10   15   20   25   30   35   40   45   50

Figure 7: Visualization of attention scores in ARCI for Ref-inpainting across different DDIM steps. We show scores from reference views attended by masked regions. The upper row shows attention scores from the 8th/16 self-attention (1/32 scale), while the bottom row shows ones from the 14th/16 self-attention (1/8 scale).

Table 3: Ablation studies for the setting of prompt tuning in Ref-inpainting. Left: 'Shallow' means only prompt tuning to text embedding, while 'Deep' indicates tuning additional embedding features to different cross-attention layers (16 layers) in SD. Right: validating the influence of the length of shared (Task) and unshared (View) prompts with 3-view Ref-inpainting.

| Prompt Type | Length | PSNR↑ | SSIM↑ | LPIPS↓ | Params |
|---|---|---|---|---|---|
| Shallow | 25 | 20.35 | 0.827 | 0.104 | +0.025M |
| Shallow | 50 | **20.49** | 0.829 | **0.103** | +0.05M |
| Shallow | 75 | 20.38 | **0.830** | 0.104 | +0.075M |
| Deep (×16) | 25(400) | 20.15 | 0.825 | 0.106 | +0.4M |

| Task | View | PSNR↑ | SSIM↑ | LPIPS↓ | Params |
|---|---|---|---|---|---|
| 50 | 0 | 21.224 | 0.838 | 0.0941 | +0.05M |
| 45 | 5 | **21.356** | **0.840** | **0.0901** | +0.06M |
| 25 | 25 | 21.127 | 0.836 | 0.0950 | +0.11M |
| 5 | 45 | 20.744 | 0.832 | 0.1040 | +0.14M |
| 0 | 50 | 20.563 | 0.831 | 0.1110 | +0.15M |

fully capture correct feature correlations without any backbone fine-tuning. As diffusion sampling progresses, self-attention modules gradually shift their focus from specific key points to broader related regions, which is convincing and intuitive. Because the key landmarks help to swiftly locate the spatial correlation between the reference and target, while the extended receptive fields further refine the generation for the following sampling steps.

**Prompt Settings.** The length and depth used in the task and view prompt tuning are explored in Tab. 3. Different from Jia et al. (2022), we find the ARCI is relatively robust in the length selection. Thus we select 50 for both Ref-inpainting and NVS. Moreover, we find that the deep prompt with much more trainable prompts for different cross-attention layers does not perform well, which may suffer from a little overfitting. For the multi-view scene, we empirically evaluate the 3-view-based Ref-inpainting performance with various proportions of task&view prompt lengths in the right of Tab. 3. Increasing the proportion of view tokens initially improves the results, followed by a subsequent decline. We think that a few unshared view tokens contribute valuable view orders, while too many unshared tokens would increase the learning difficulty, leading to an inferior prompt tuning performance.

Table 4: Results of NVS with different reference views with/without incremental Positional Encoding (PE).

| Ref-View | PE | PSNR↑ | SSIM↑ | LPIPS↓ |
|---|---|---|---|---|
| 1 | ✗ | 20.352 | 0.873 | 0.132 |
| 1 | ✓ | **20.508** | **0.875** | **0.128** |
| 4 | ✗ | 22.097 | 0.888 | 0.099 |
| 4 | ✓ | **22.324** | **0.890** | **0.095** |

**Incremental Positional Encoding.** We incrementally add the concatenation of learnable view embedding and absolute positional encoding to each attention block for NVS (Eq. 1), improving the performance of both single-view and multi-view-based NVS as verified in Tab. 4. More ablation studies are verified in the Appendix. A.4.1.

## 5 CONCLUSION

In this paper, we propose ARCI, formulating reference-based multi-view image synthesis as inpainting tasks and addressing them end-to-end. Benefiting from the prompt tuning and the well-learned attention modules in large T2I models, ARCI can address the spatially sophisticated Ref-inpainting and NVS efficiently. Moreover, ARCI could be easily extended to tackle multi-view generation tasks. We also propose block casual masking to accomplish NVS with consistent results autoregressively. Comprehensive experiments on Ref-inpainting and NVS show the effectiveness of ARCI.

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

# A    APPENDIX

## A.1    BROADER IMPACTS

This paper exploited image synthesis with text-to-image models. Because of their impressive generative abilities, these models may produce misinformation or fake images. So we sincerely remind users to pay attention to it. Besides, privacy and consent also become important considerations, as generative models are often trained on large-scale data. Furthermore, generative models may perpetuate biases present in the training data, leading to unfair outcomes. Therefore, we recommend users be responsible and inclusive while using these text-to-image generative models. Note that our method only focuses on technical aspects. Both images and pre-trained models used in this paper are all open-released.

## A.2    DATA PROCESSING AND OTHER IMPLEMENTATION DETAILS

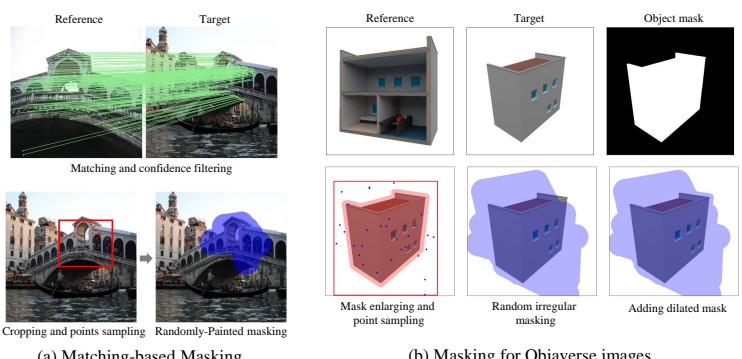

Figure 8: The illustration of (a) matching-based masking for Ref-inpainting, and (b) masking strategy used for NVS on Objaverse (Deitke et al., 2022).

### A.2.1    MATCHING-BASED MASKING AND DATA PROCESSING FOR REF-INPAINTING

For the Ref-inpainting, we find that the widely used irregular mask (Dong et al., 2022; Zhou et al., 2021; Zhao et al., 2022b) fails to reliably evaluate the capability of spatial transformation and structural preserving. Therefore, as shown in Fig. 8(a), we propose the matching-based masking method. Specifically, we first utilize the scene info provided by MegaDepth (Li & Snavely, 2018) to select out the image pairs which have an overlap rate between 40% and 70% Second, for each image pair, we use a feature matching model (Tang et al., 2022) to detect matching key-points between the images and assign each key-points pair a confidence score. Next, we filter out those pairs with low confidence scores with the threshold of 0.8. Then we randomly crop a 20% to 50% sub-space in the matched region and sample 15 to 30 key points as vertices to be painted across for the final masks. The matching-based mask not only improves the reliability during the evaluation but also facilitates the performance in the training phase as in Tab. 6.

We split 505 pairs from MegaDepth (Li & Snavely, 2018) as the validation, including some manual masks from ETH3D scenes (Schops et al., 2017). For the multi-view testing set, we further filter all scenes and retain the ones with at least 4 reference views. Thus there are 482 images in the final multi-view testing set.

Table 5: Training details of ARCI. NVS (4-view) and Ref-inpainting (4-view) are trained on ×8 and ×4 A800 GPUs respectively, while others are trained on ×2 A6000 GPUs. NVS (4-view) is fine-tuned based on NVS (1-view).

| Task | Batch size | Learning rate | | Steps | Training time |
| --- | --- | --- | --- | --- | --- |
| | | Prompt&LoRA | Backbone | | |
| Ref-inpainting (1-view) | 16 | 3e-5 | / | 6k | 9h |
| Ref-inpainting (2-view) | 16 | 3e-5 | / | 6k | 10h |
| Ref-inpainting (3-view) | 24 | 3e-5 | / | 16k | 14h |
| Ref-inpainting (4-view) | 64 | 5e-5 | / | 16k | 12h |
| NVS (1-view) | 48 | 1e-4 | 1e-5 | 80k | 34h |
| NVS (4-view) | 512 | 1e-4 | 3e-5 | 40k | 62h |

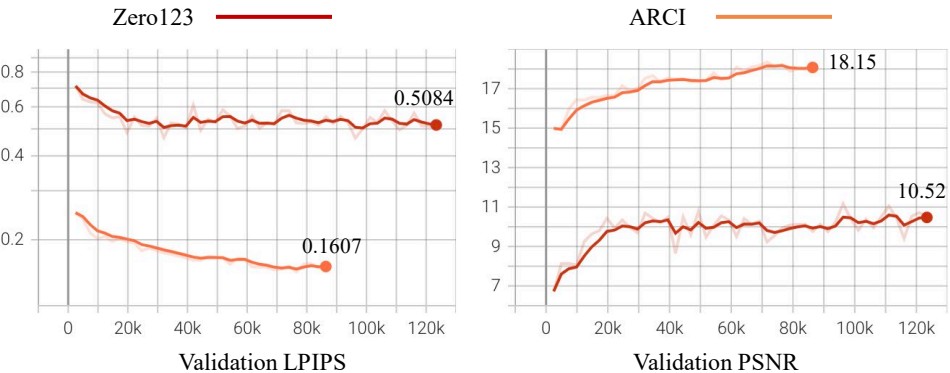

Figure 9: Training logs of ARCI and Zero123 (Liu et al., 2023b) of the NVS on Objaverse (Deitke et al., 2022) with the setting of batch size 48 and learning rate 1e-5.

### A.2.2 Data Processing for NVS

For the NVS, we first dilate the object mask and randomly sample points in the enlarged mask bounding box to paint the irregular mask. Then, we unite the dilated object mask to completely cover target images as in Fig. 8(b). We find that local masking is still very important for fast convergence and stable fine-tuning as empirically verified in experiments. For the data processing on Objaverse (Deitke et al., 2022), Zero123 (Liu et al., 2023b) provided images including 800k various scenes with object masks. For each scene, 12 images are rendered in 256×256 with different viewpoints. Following Liu et al. (2023b), the spherical coordinate system is used to convert the relative pose $\Delta p$ into the polar angle $\theta$, azimuth angle $\phi$, and radius $r$ distanced from the canonical center as $\Delta p = (\Delta\theta, \sin\Delta\phi, \cos\Delta\phi, \Delta r)$, where the azimuth angle is sinusoidally encoded to address the non-continuity. For the masking of Objaverse images, we dilate the object mask and related bounding box with 10 to 25 kernel size and 5% to 20% respectively. Then we randomly sample 20 to 45 points to paint the irregular masks.

We select 500 scenes from Objaverse as the validation, while others are used as the training set. Note that there exists an overlap between our validation and Zero123's training set (Liu et al., 2023b), but our method still outperforms the official Zero123 as in Tab. 2.

### A.2.3 Training Details

We show the training details in Tab. 5. ARCI is efficient in being trained for various tasks. To further demonstrate the effectiveness of ARCI, we provide the training log of ARCI and Zero123 in Fig. 9. Obviously, the contextual inpainting-based ARCI enjoys a substantially faster convergence and superior performance.

### A.3 Autoregressively Sequential Generation

To verify the generalization of our method, we generate more groups of multi-view images through a single input view as in Fig. 10. Moreover, we test several real-world cases with one RGB input in

---

**Algorithm 1** Pseudo codes for block casual masking.

```
# view: the view number
# length: length of the sequence, usually be h*w

mask = zeros((view, length)) # [view,length]
mask[:, 0] = 1
mask = cumsum(mask.reshape(1, view * length), dim=1) # [1,view*length]
mask = (mask.T >= mask).float()  # [view*length,view*length]
mask = 1 - mask # masked regions are 1, unmasked regions are 0
mask = mask.masked_fill(mask == 1, -inf) # let all masked regions to -inf
```

---

**Algorithm 2** Pseudo codes for the attention visualization.

```
# x: [b,2hw,c], input feature for attention module (left:reference, right:target)
# mask: [b,2hw,1], input 0-1 mask; 1 means masked regions

q, k = matmul(x, Wq), matmul(x, Wk) # [b,2hw,c], project x to query (q) and key (k)
A = matmul(q, k.T) # [b,2hw,2hw], get attenion map
A = mean(A * mask , dim=1) # [b,2hw] get mean scores attended by masked regions
A = A.reshape(b,h,w)[:, :, :w//2] # [b,h,w], show reference attention score only
```

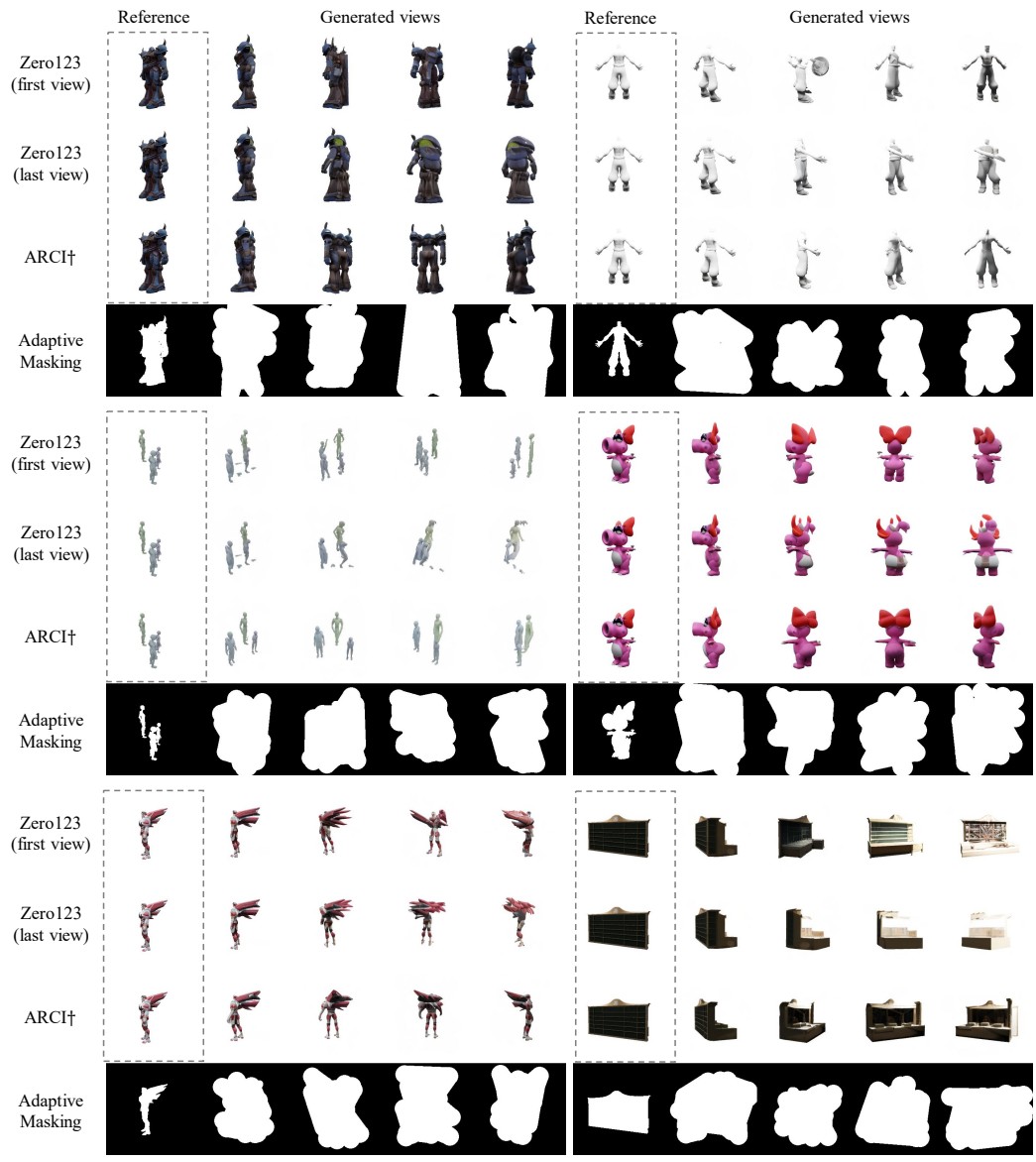

Figure 10: The sequential generative results from a single view. Zero123's (Liu et al., 2023b) results are conditioned on the real reference (first view) and the last generated view (last view) respectively.

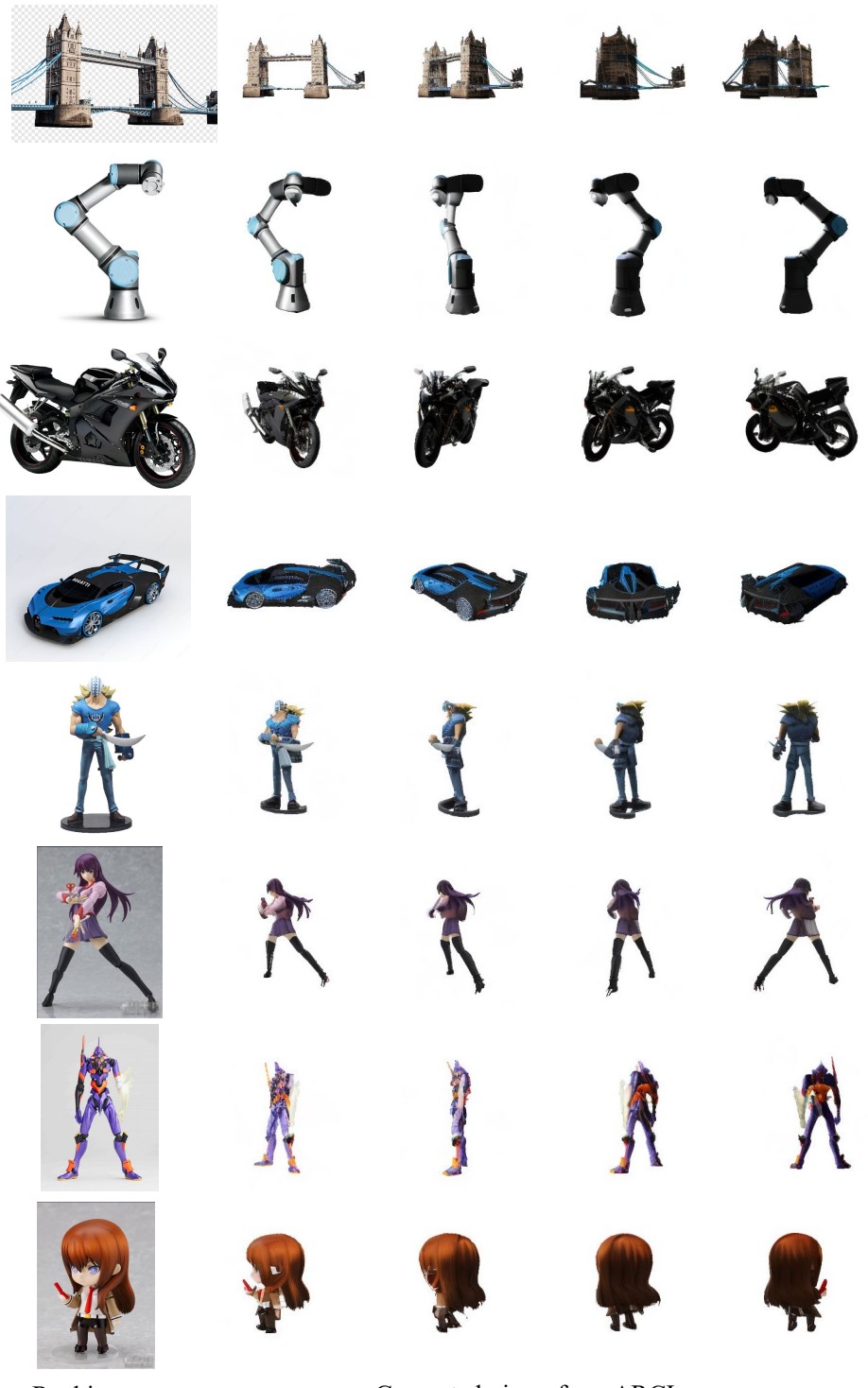

Real image                Generated views from ARCI

Figure 11: Consistent real-world NVS results generated by ARCI.

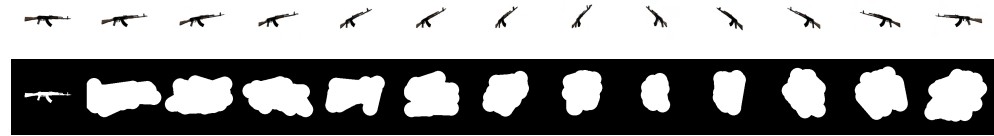

Figure 12: Long sequence synthesis from a single image (upper) with adaptive masking (bottom). The leftmost image and mask are the input while others are generated.

Fig. 11. All poses are initialized to $[0.5\pi, 0, 1.5]$ for polar angle, azimuth angle, and radius distance, respectively. The proposed ARCI can be well generalized to real-world cases.

### A.3.1 ADAPTIVE MASKING

One may ask that the masking strategy used in Fig. 8(b) suffers from some shape leakages, which lead to unreliable metrics in Tab. 2. We should clarify that our method can perform well only with the reference mask, which is easy to get by the salient object detection (Qin et al., 2020). Specifically, we dilate the reference mask as Fig. 8(c). Then, a few DDIM steps (Song et al., 2020) are used to generate a rough synthesis in the target view. After that, we detect the foreground mask based on the rough synthesis by Qin et al. (2020) and further dilate this mask for the second synthesis with full DDIM steps. The adaptive masking can be well generalized to the NVS as verified in Fig. 12. All testing results in this paper are already based on adaptive masking. Besides, we think that providing target masks according to the distance and direction priors manually is also convincing to address the challenging single-view-based NVS.

## A.4 SUPPLEMENTAL EXPERIMENTAL RESULTS

### A.4.1 SUPPLEMENTAL ABLATION STUDIES

Table 6: Ablation studies of Ref-inpainting on MegaDepth. Left: effects of matching-based masks and inference noise $\eta$. Right: effects of different prompt initialization.

| Configuration | PSNR↑ | SSIM↑ | LPIPS↓ | Prompt init | PSNR↑ | SSIM↑ | LPIPS↓ |
|---|---|---|---|---|---|---|---|
| baseline | 20.489 | 0.829 | 0.1029 | Random | 20.810 | 0.832 | 0.0998 |
| + Match mask | 20.574 | 0.830 | 0.1010 | Token-wise | 20.852 | 0.833 | 0.1002 |
| + $\eta$=1.0 | **20.993** | **0.837** | **0.0951** | Token-avgs | **20.926** | **0.836** | **0.0961** |

**Matching-based Masks and Noise Coefficient.** On the left of Tab. 6, we find that the matching-based mask enjoys substantial improvement in the reference-guided inpainting. Besides, setting the noise coefficient $\eta = 1$ achieves consistent improvements in our ARCI even sampled as the DDIM (Song et al., 2020). So all LDMs are worked under $\eta = 1$ without special illustrations.

**Prompt Initialization.** We tried three initialization ways for the prompt tuning on the right of Tab. 6. The random initialization performs worst. Both 'token-wise' and 'token-avgs' leverage text embeddings from a task-specific descriptive sentence that is detailed in the supplementary. 'Token-wise' means repeating descriptive sentences until the prompt length, while each token is initialized for one prompt token. 'Token-avgs' indicates that all prompt tokens are initialized with the average of the descriptive sentence. Meaningful initialization is useful for task-specific prompt tuning.

Table 7: Abaltions of CFG on Objaverse (Deitke et al., 2022) NVS.

| CFG training | CFG weight | PSNR↑ | SSIM↑ | LPIPS↓ |
|---|---|---|---|---|
| × | 1.0 | 20.310 | 0.872 | 0.1318 |
| ✓ | 1.0 | 20.352 | 0.873 | 0.1322 |
| ✓ | 1.5 | **20.528** | 0.874 | 0.1297 |
| ✓ | 2.5 | 20.508 | **0.875** | **0.1288** |
| ✓ | 5.0 | 20.077 | 0.873 | 0.1310 |

Table 8: Abaltions of CFG on MegaDepth (Li & Snavely, 2018) Ref-inpainting.

| Ref Views | CFG weight | PSNR↑ | SSIM↑ | LPIPS↓ |
|---|---|---|---|---|
| 1 | 1.0 | **21.502** | **0.840** | 0.1030 |
| | 1.5 | 21.482 | **0.840** | 0.0955 |
| | 2.0 | 21.195 | 0.837 | **0.0946** |
| | 2.5 | 20.761 | 0.832 | 0.0969 |
| 2 | 1.0 | **21.511** | **0.840** | 0.105 |
| | 1.5 | 21.451 | **0.840** | 0.0977 |
| | 2.0 | 21.092 | 0.836 | **0.0969** |
| | 2.5 | 20.614 | 0.830 | 0.0997 |
| 3 | 1.0 | **21.771** | **0.844** | 0.0991 |
| | 1.5 | 21.703 | **0.844** | 0.0912 |
| | 2.0 | 21.356 | 0.840 | **0.0901** |
| | 2.5 | 20.855 | 0.834 | 0.0929 |
| 4 | 1.0 | **22.334** | **0.851** | 0.0902 |
| | 1.5 | 22.197 | 0.851 | **0.0836** |
| | 2.0 | 21.779 | 0.847 | 0.0839 |
| | 2.5 | 21.125 | 0.8407 | 0.0894 |

Table 9: Ref-inpainting results on the real-world set provided by Zhou et al. (2021).

| Method | PSNR↑ | SSIM↑ | FID↓ | LPIPS↓ |
|---|---|---|---|---|
| ProFill (Zeng et al., 2020) | 25.550 | 0.944 | 71.758 | 0.0848 |
| TransFill (Zhou et al., 2021) | **26.052** | **0.945** | 62.493 | 0.0757 |
| **ARCI** | 25.733 | 0.942 | **61.276** | **0.0756** |

**Effectiveness of CFG.** We remove the pose condition with 15% to train the ARCI for NVS. Then the CFG coefficient 2.5 is used during the inference. As verified in Tab. 7, CFG could improve the performance with better pose control. Moreover, we find that CFG can also enhance the performance of Ref-inpainting even without training with prompts dropout as in Tab. 8. The LPIPS first decreases, but then increases as the CFG decreasing from 2.5 to 1.0, while the PSNR and the SSIM keep increasing. We regard LPIPS as the most important metric, which conforms to the human perception. Therefore, we set CFG to 2.0 when testing our model for Ref-inpainting.

### A.4.2 RESULTS OF REF-INPAINTING

We provide more qualitative and quantitative results of Ref-inpainting[2] in Fig. 13, Fig. 14, and Tab. 9. We further provide qualitatively multi-view Ref-inpainting results in Fig. 15

### A.4.3 RESULTS OF NVS

Besides, we show some diverse NVS on Objaverse (Deitke et al., 2022) in Fig. 16. Different random seeds are utilized to process the DDIM sampling. ARCI can achieve reasonable results with correct target poses. More results are in Fig. 17 and Fig. 18.

### A.5 INFERENCE SPEED

We provide the inference speed for different input resolutions in Tab. 10. All tests are based on one 32GB V100 GPU with 50 DDIM steps. ARCI needs to stitch two images together, which would double the input size. But the inference time is not doubled as shown in Tab. 10. Note that when the image size is smaller than 512, the difference in inference costs is not obvious.

Table 10: Inference speed of SD under 50 DDIM sampling steps.

| Input | sec/img |
|---|---|
| 256×256 | 2.9172 |
| 256×512 | 2.9395 |
| 512×512 | 3.0715 |
| 512×1024 | 4.0205 |

---

[2]Since TransFill (Zhou et al., 2021) is not released, we send our images and masks to the authors and take their inpainted results for the evaluation.

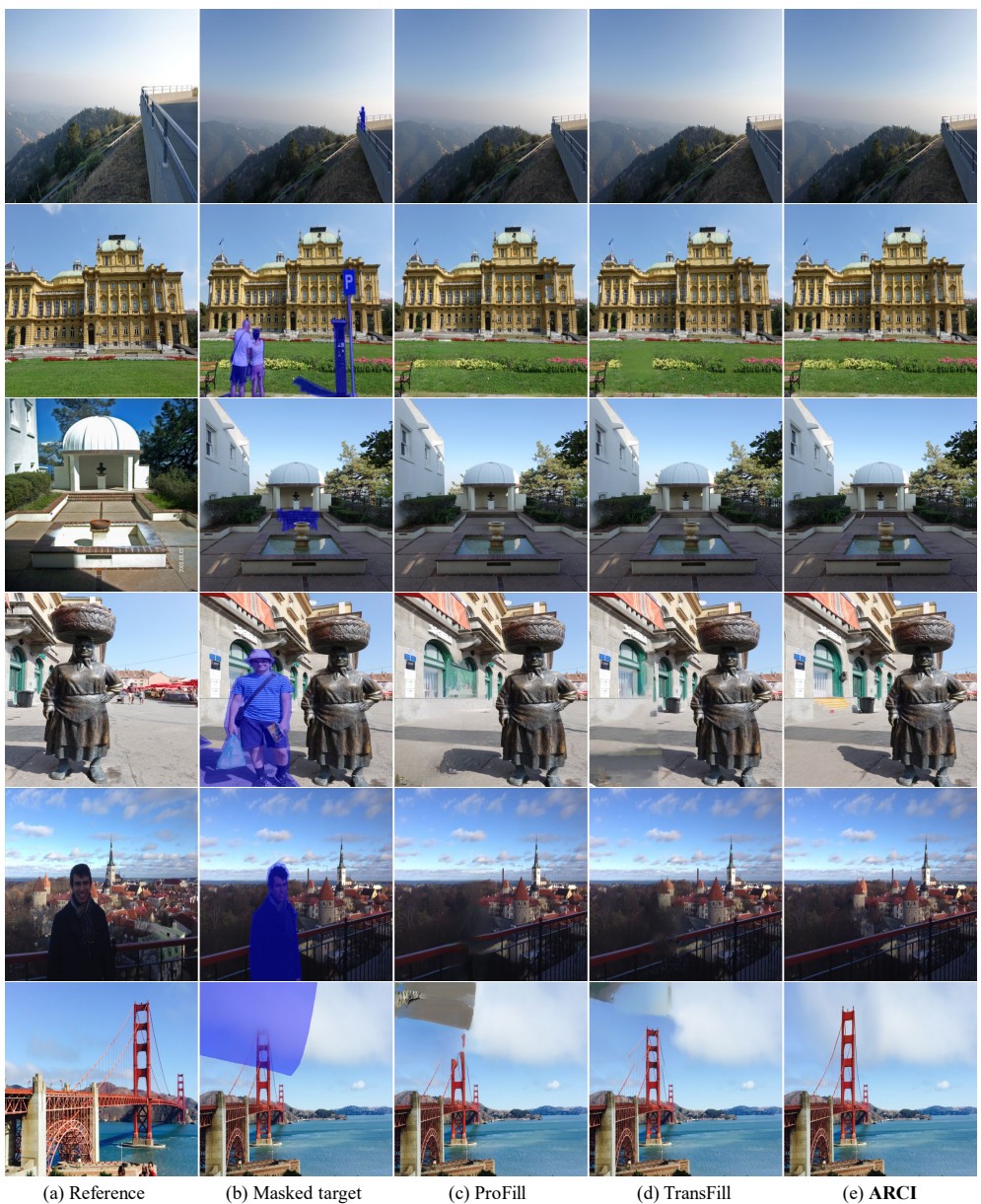

(a) Reference      (b) Masked target      (c) ProFill      (d) TransFill      (e) **ARCI**

Figure 13: Qualitative Ref-inpainting results compared with ProFill (Zeng et al., 2020), TransFill (Zhou et al., 2021), ARCI on the challenging real set provided by TransFill (Zhou et al., 2021).

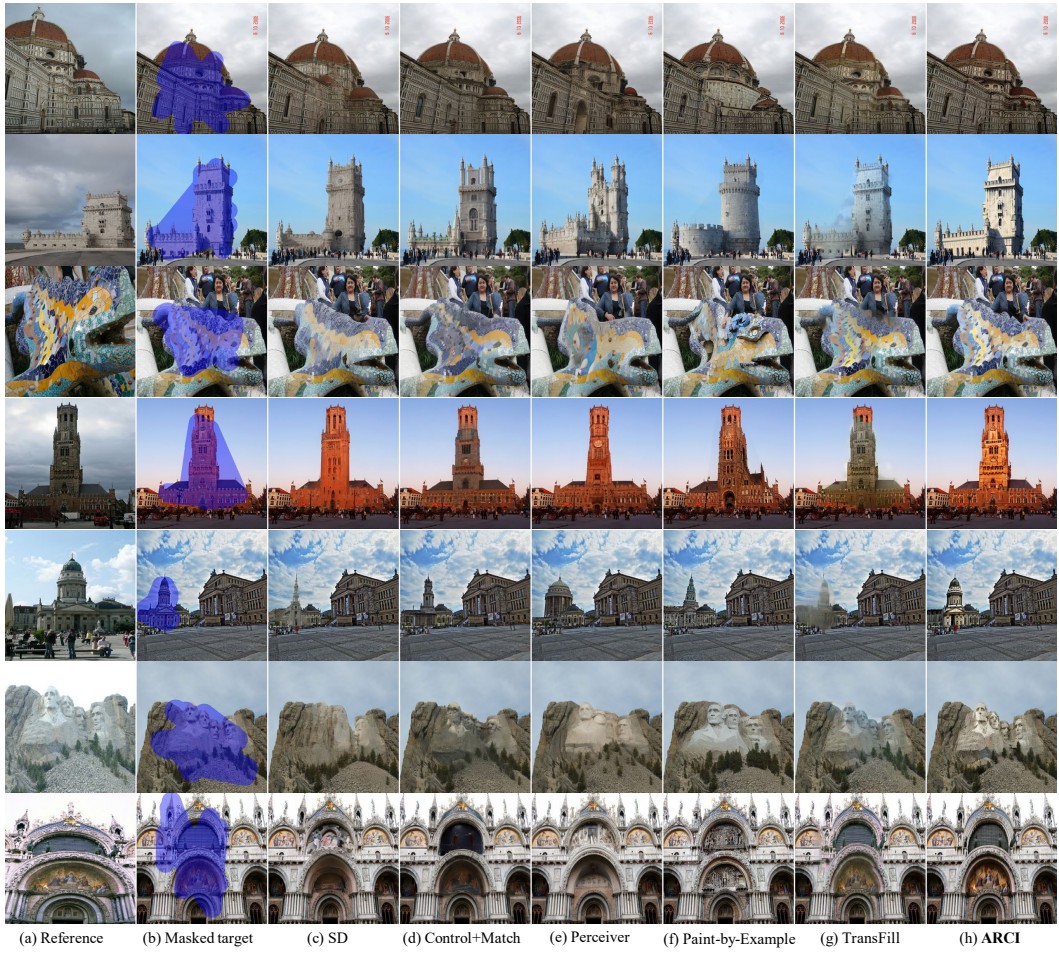

(a) Reference   (b) Masked target   (c) SD   (d) Control+Match   (e) Perceiver   (f) Paint-by-Example   (g) TransFill   (h) **ARCI**

Figure 14: Qualitative Ref-inpainting results on MegaDepth, which are compared among (c) SD (Rombach et al., 2022), (d) ControlNet (Zhang & Agrawala, 2023)+Matching (Tang et al., 2022), (e) Perceiver (Jaegle et al., 2021) with ImageCLIP (Radford et al., 2021), (f) Paint-by-Example (Yang et al., 2023), (g) TransFill (Zhou et al., 2021), and (I) our ARCI. Please zoom in for more details.

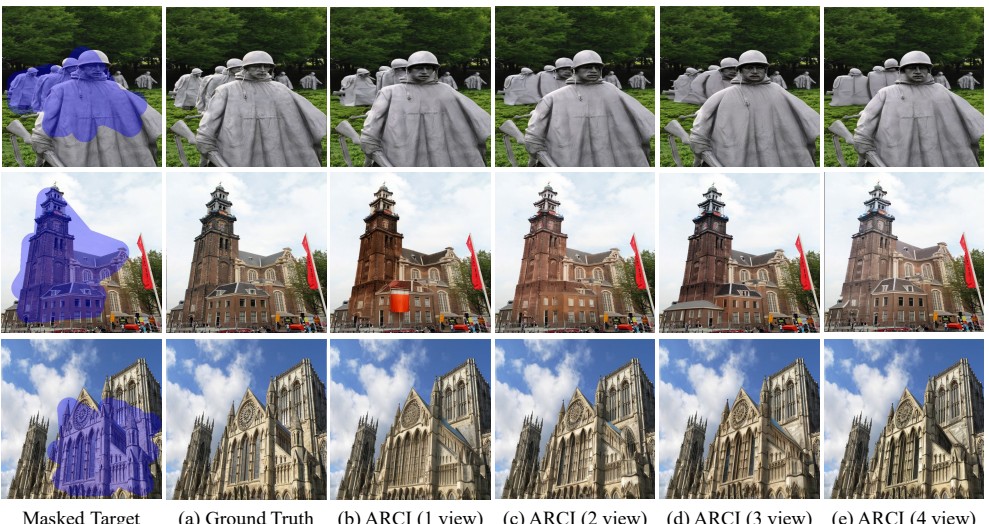

Masked Target   (a) Ground Truth   (b) ARCI (1 view)   (c) ARCI (2 view)   (d) ARCI (3 view)   (e) ARCI (4 view)

Figure 15: Multi-view Ref-inpainting qualitative results. Increasing the reference view number improves the image quality of repaired targets.

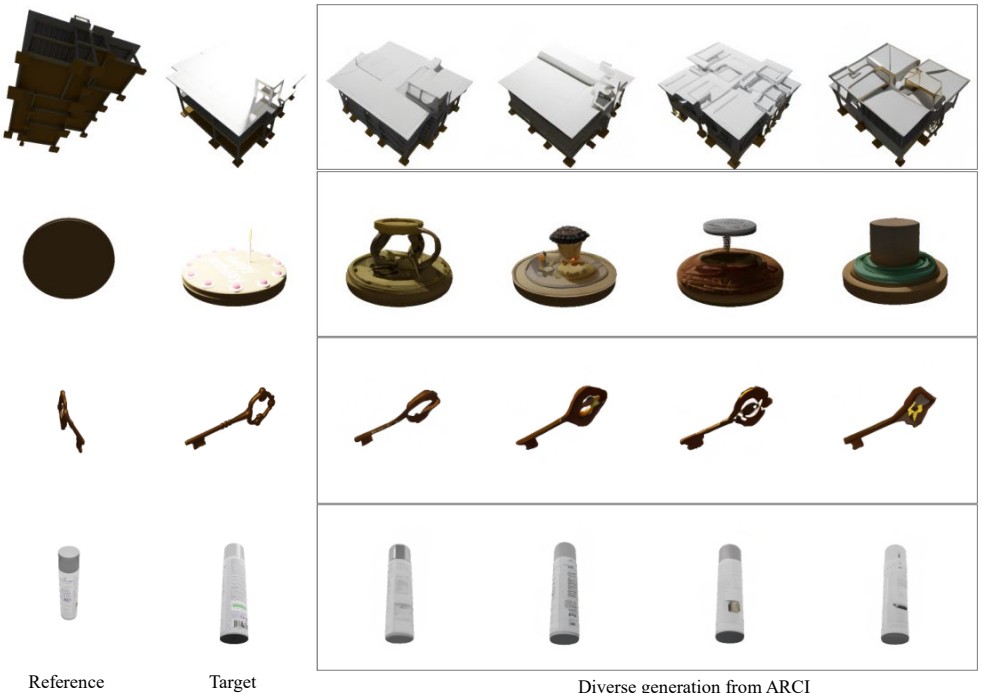

Reference      Target          Diverse generation from ARCI

Figure 16: Diversity of the NVS on Objaverse (Deitke et al., 2022) from a single reference image without multi-view guidance.

Therefore, we think that the inference cost of the proposed ARCI is still acceptable in most real-world applications.

### A.6 USER STUDY

To evaluate the effectiveness of our ARCI in Ref-inpainting. We further test the user study as the human perceptual metric in Fig. 19. Formally, 50 masked image pairs are randomly selected from our test set which are compared among SD (Rombach et al., 2022), ControlNet (Zhang & Agrawala, 2023)+match (Tang et al., 2022), Perciver (Jaegle et al., 2021), Paint-by-Example (Yang et al., 2023), TransFill (Zhou et al., 2021), and ARCI. Although TransFill was not open-released, we thank TransFill's authors for kindly testing these samples for us. There are 10 volunteers who are not familiar with image generation attending this study. Given masked target images and reference ones, we ask volunteers to vote for the best recovery from the 6 competitors mentioned above. The voting criterion should consider both the faithful recovery according to the reference and natural generations of color and texture. As shown in Fig. 19, ARCI outperforms other competitors.

### A.7 LIMITATION

Although the proposed ARCI enjoys good performance and geometric consistency in multi-view NVS, it still suffers from the drawback of *error accumulation* as shown in Fig. 20. To eliminate this problem, we recommend providing a few more views (2,3,4) for more robust geometric priors.

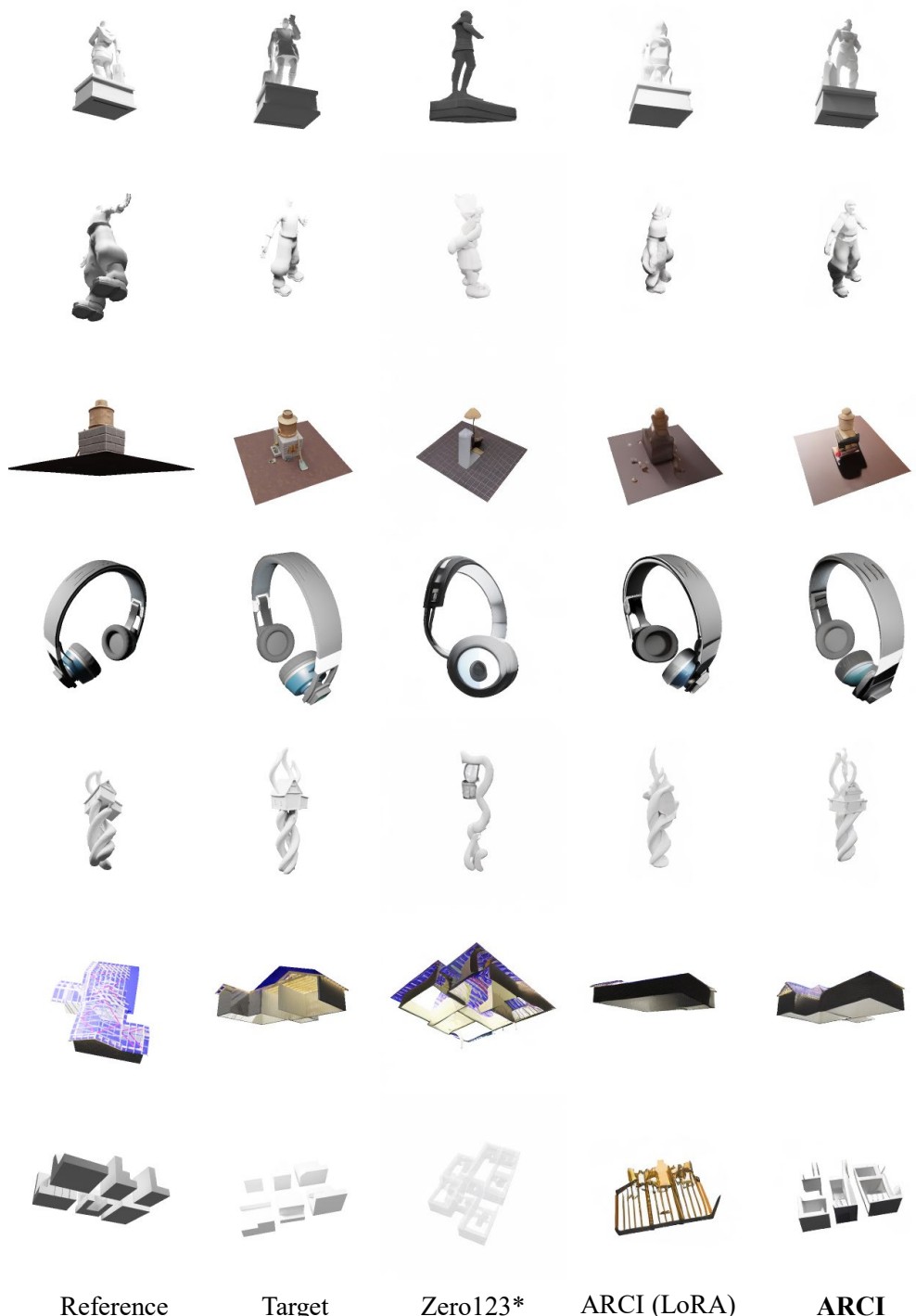

Reference      Target      Zero123*      ARCI (LoRA)      **ARCI**

Figure 17: NVS on Objaverse (Deitke et al., 2022) from a single reference image.

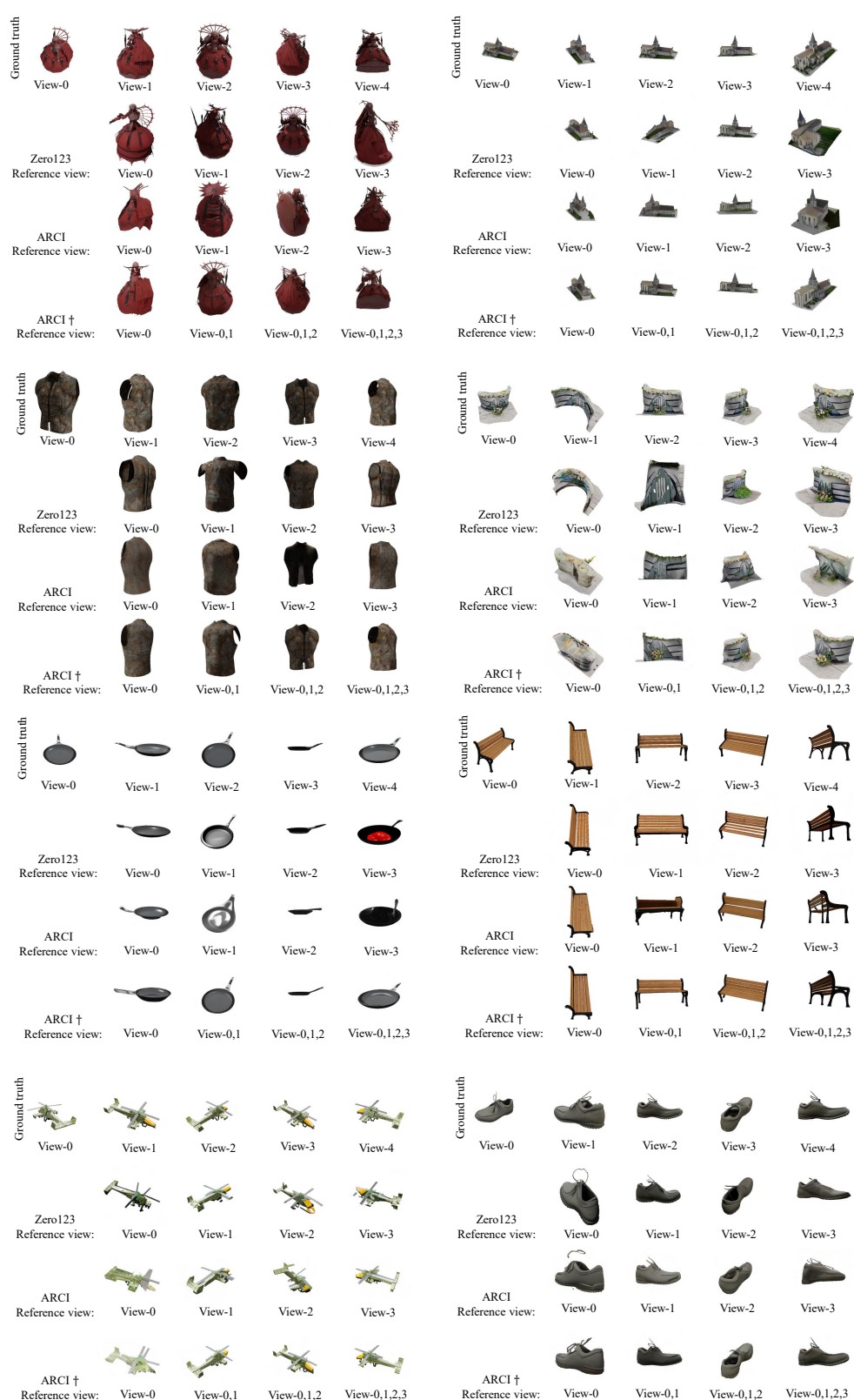

Figure 18: Multi-view NVS results on Objaverse compared among the official Zero123 (Liu et al., 2023b), one-view based ARCI, and multi-view based ARCI† trained with 512 batch size. Please zoom in for details.

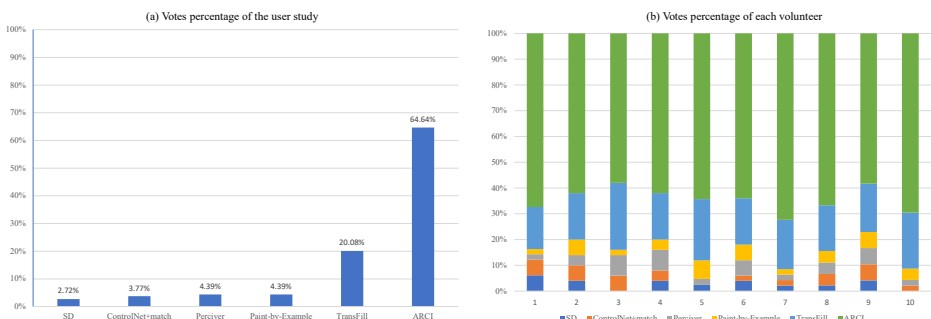

Figure 19: The user study evaluation; (a) the overall voting percentage; (b) the votes of each volunteer.

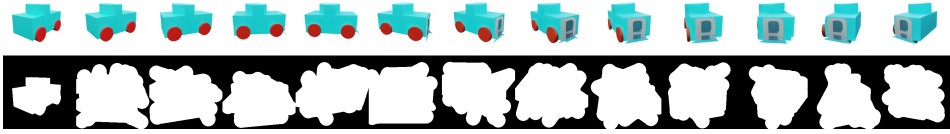

Figure 20: The error accumulation occurred in AR generation. The degraded result is first generated in view 3.

