# OpenReview forum: "Harnessing Attention Prior for Reference-based Multi-view Image Synthesis"
_ICLR.cc/2024/Conference — ICLR 2024 Conference Withdrawn Submission_

### Official Review · Reviewer_RjKx · 2023-10-28

**Soundness:** 3 good
**Presentation:** 3 good
**Contribution:** 3 good
**Rating:** 6
**Confidence:** 3

**Summary:**

This paper proposes an approach for reference-based multi-view synthesis that supports both image inpainting from reference samples and novel view synthesis of the reference image. They are all formulated as contextual inpainting tasks. The proposed ARCI enhances attention mechanisms in T2I models by learning correlations across different reference view with self attention and control novel view synthesis with cross attention. Both qualitative and quantitative experiments have been conducted to demonstrate the effectiveness of the proposed method.

**Strengths:**

1. The paper regards multi-view image synthesis as reference-based inpainting task, which is novel and provides an interesting direction for realizing NVS.

2. The results of novel view synthesis are excellent.

3. The proposed Block Casual Masking bridges the gap of converting a diffusion model to a AR-based generative model.

**Weaknesses:**

The paper needs to improve readability by structuring the content more logically.

**Questions:**

1. Do view embeddings require retraining for each image?

2. For multi-view NVS task, does the LDM need fintuning for each new image tested?

3. How to control the view direction of multi-view synthesis?

---

> ### Author Response · Authors · 2023-11-16
> **Response to Reviewer RjKx**
>
> Thanks for your valuable feedback. We would further improve the readability in our revision.
>
> **1. Do view embeddings require retraining for each image?**
>
> Thanks. Our ARCI is a generalized model without any test-time fine-tuning for both Ref-inpainting and NVS.
>
> **2. For multi-view NVS task, does the LDM need fintuning for each new image tested?**
>
> Thanks. Benefiting from the autoregressive training of ARCI, all views can be generalized with the same model without test-time fine-tuning.
>
> **3. How to control the view direction of multi-view synthesis?**
>
> Thanks. For multi-view NVS, we provide a relative pose for each view (based on the first view) to pose FC as in Fig.2(c).

---

### Official Review · Reviewer_d1up · 2023-10-29

**Soundness:** 2 fair
**Presentation:** 2 fair
**Contribution:** 2 fair
**Rating:** 3
**Confidence:** 5

**Summary:**

This paper focuses on the task of novel view image synthesis. The authors introduce Attention Reactivated Contextual Inpainting (ARCI) technique for both reference-based inpainting and novel view synthesis. The authors show comparison results on MegaDepth, ETH3D and Objaverse.

**Strengths:**

1. This paper conducts many experiments and shows extensive quantitative and qualitative comparisons for different applications in novel view synthesis.
2. The authors show some good results in reference-based image inpainting.

**Weaknesses:**

1. The definition is a bit confusing. Why do we need the concept of local synthesis and global synthesis for novel view synthesis? What’s the core challenge? Besides, the claimed local synthesis is actually the reference-guided inpainting task, and the proposed ARCI is built upon an inpainting model, which makes the target more like inpainting.

2. The presentation is not clear enough. for example, in Figure 1, the authors should explain what the purple mask means. What is the input of the model, the green bounding box or the purple one? The inputs and outputs look very different in (a)-(d).

3. The authors claim in the introduction that they can use efficient task and view prompt tuning for novel view synthesis with frozen SD, However, in the experiments, the authors finetuned the whole stable diffusion and reported some results with Lora and fully finetuned results. I think the authors should grandly review and improve their presentation and make them clearer.

4. The comparison in Figure 5 is not fair. The authors should report the original results of Zero123 instead of re-training it with a suboptimal training setting. Besides, zero123 is able to take specific camera poses to generate the novel view, what’s the input used here for both zero123 and ARCI? The authors should make sure all the models take appropriate inputs.

5. Some important experiments are missing. The authors should conduct an ablation study on adaptive masking. For comparison with zero-123 for novel view synthesis, the authors should show comparisons on GSO and RTMV following zero-123. For reference-based inpainting, the authors should also show comparison results by using some very different references following the setting of Paint-by-Example to verify the effectiveness of the ref-inpainting technique for open-domain cases.

6. What’s the used task and view prompt? Specifically, do the task prompts indicate local or global? How to set the view prompts? Rotation angles or front/side/back view? If the model tasks as input an image beyond these views, then how does the ARCI work?

**Questions:**

It is a bit hard for me to fully understand this paper. It looks like there are two different tasks (without much analysis and relations) in the same paper, i.e., reference-based inpainting and novel view synthesis. Besides, instead of following existing standard benchmarks, the authors design new comparison settings for different tasks in this paper. Many details are missing especially about the detailed setting of different tasks and view prompts. For example, how to correlate the concrete prompt with concrete views in the training dataset?  Please find more details in the Weaknesses and address my concerns.

---

> ### Author Response · Authors · 2023-11-16
> **Response to Reviewer d1up**
>
> Thanks for your valuable feedback. We recognize that the pivotal contribution of our work has not been adequately emphasized and will make a major revision to our paper to rectify this issue.
>
> **1. Confused definition for two tasks.**
>
> Thanks. We would definitely re-write our paper to improve the presentation and definition.
>
> **2. Unclear presentation for Fig.1**
>
> Thanks. We would improve the presentation and redraw Fig.1.
>
> **3. Why does ARCI with prompt tuning need fine-tuning?**
>
> Thanks for this good point! We have to clarify that ARCI based on **Ref-inpainting does not require any fine-tuning for LDM**, while NVS requires fine-tuning for superior results, which have been illustrated in Fig.2 and Sec.3.
> Moreover, our method also enjoys much faster convergence compared to previous methods like Zero123 as we verified in Fig9 of the Appendix.
> Besides, both ARCIs for Ref-inpainting and NVS are generalized models, without requiring any test-time fine-tuning.
>
> **4. Unfair comparison of Fig5. What's the input of Zero123 and ARCI?**
>
> Thanks. We would add the results of official zero123 to Fig5 in the revision. Moreover, inputs for ARCI and Zero123 are the same, including a reference image and relative poses.
>
> **5. Missing experiments for NVS and very different references for Ref-inpainting.**
>
> Thanks. We have compared our method to Zero123 in GSO dataset in the Appendix. For the very different references in Ref-inpainting, we have to clarify that this is not our goal in reference-guided inpainting, which has been clearly defined in Sec.2.3 (all reference and target views should be taken from the same object from different viewpoints).
>
> **6. Details about the prompts used in ARCI**
>
> For  NVS, we follow Zero123 to encode polar, azimuth, and radius distance which are detailed in the Appendix.
> We use shared (task) and unshared (view) trainable embeddings as task and view prompt, which are encoded by CLIP-H and further infused into LDM through cross-attention (Fig.2). We will further claim these in our revision.

---

### Official Review · Reviewer_frhM · 2023-10-29

**Soundness:** 2 fair
**Presentation:** 1 poor
**Contribution:** 2 fair
**Rating:** 3
**Confidence:** 5

**Summary:**

The paper proposes a method to generate multi-view images. The major novelty is a combination of local synthesis and global synthesis.

**Strengths:**

1. The results seem to be good. The numerical results indicate the method to achieve state-of-to-the-art performance.
2. The proposed methods are general. It can be applied to multiple tasks, e.g. single-view NVS, multi-view NVS.

**Weaknesses:**

1. The writing is too bad. The paper has many confused words, unclear explanation, and unconvincing arguments. Here are some examples:
    * Confused words. In abstract, "Our contributions of ARCI, built upon the Stable Diffusion fine-tuned for text-guided inpainting, include skillfully handling difficult multi-view synthesis tasks with off-the-shelf T2I models, introducing task and view-specific prompt tuning for generative control, achieving end-to-end Ref-inpainting, and implementing block causal masking for autoregressive NVS. " The sentences indicates there are 3 contribution: 1) handling difficult multi-view synthesis tasks with off-the-shelf T2I models, 2) introducing task and view-specific prompt tuning for generative control, achieving end-to-end Ref-inpainting, 3) Implementing block causal masking for autoregressive NVS. The phrase 'block causal masking' is first introduced here, which confuses readers. In addition, why implementing this can be counted as a contribution? I suggest the author explain what the block causal masking is first.
    * Unclear explanation. In introduction, "This task can be broadly categorized into two facets: local and global multi-view image synthesis from reference images. " If I understand Figure 1 correctly, there seems to be 2 tasks, ref-impainting and novel view synthesis here, but the author write one task with 2 facets. I think this is a confusing definition.
   * Unconvincing arguments. In introduction, 'They struggle to capture fine-grained correlations, including object orientations and precise object locations, between reference and target images. These nuanced details are pivotal for tasks such as multi-view generation, as exemplified by Ref-inpainting.' Is there any reference paper or experimental results supporting this arguments?
The above shows 3 typical examples and there are many more here and there. The paper writing does not satifiy ICLR bar and requires major re-writing.
2. What's the motivation of this paper? 'However, adapting them for multi-view synthesis is challenging due to the intricate correlations between reference and target images.' This sentence is correct, this is not the problem of current methods, e.g. Zero-1-to-3. Why does this proposed method better than zero-1-to-3. I suggest the author to write the motivation clearly in abstract.
3. In related work, 'Compared with these aforementioned manners, the proposed ARCI enjoys both spatial modeling capability and computational efficiency.' What is spatial modeling capability? Please define spatial modeling capability.
4. In method, please define and formulate the two tasks first.
5. ATTENTION REACTIVATED CONTEXTUAL INPAINTING is not intuitive. It's hard to image the technique by the name. I recommentd to use more intuitive method name.
6. The method design seems to be complicated or the method writing is bad, which make the method design look complicated.

**Questions:**

I have many questions, but many of them are due to unclear writing. I recommend the author to re-write the entire paper and I can re-rate the paper.

---

> ### Author Response · Authors · 2023-11-16
> **Response to Reviewer frhM**
>
> Thanks for your valuable feedback. We recognize that the pivotal contribution of our work has not been adequately emphasized and will make a major revision to our paper to rectify this issue.
>
> **1. Confused and unclear explanation and motivation**
>
> Thanks. We would definitely re-write our paper to improve the presentation.
>
> **2. Unconvincing arguments: "They struggle to capture fine-grained correlations" and "spatial modeling capability"**
>
> Thanks for this good point. We have to clarify that we have proved these in the qualitative and quantitative experiments in Fig.4 and Tab.1.
> For Ref-inpainting, which needs to learn spatial relations, previous methods like ControlNet fail to address it properly.
> Thus our claims that ARCI enjoys good fine-grained correlations and spatial modeling capability are reasonable.
>
> **3. Please define and formulate the two tasks first in method**
>
> Thanks. We would clearly define two tasks at the start of the method (Sec.3).
>
> **4. ATTENTION REACTIVATED CONTEXTUAL INPAINTING is not intuitive**
>
> Thanks. We would change to a new name for our method.

---

### Official Review · Reviewer_a7pP · 2023-10-31

**Soundness:** 2 fair
**Presentation:** 1 poor
**Contribution:** 2 fair
**Rating:** 3
**Confidence:** 4

**Summary:**

This paper proposes Attention Reactivated Contextual Inpainting (ARCI), a unified method to leverage pre-trained text-to-image diffusion models (e.g., Stable Diffusion) to complete inpainting tasks. The key ideas of ARCI are i) using the attention layers in T2I diffusion models to learn correlations across different views and ii) using cross-attention layers to inject extra control via prompt tuning. Experiments are conducted on Ref-inpainting (local inpainting) and novel view synthesis (global inpainting). ARCI shows superior performance on both tasks compared to existing methods and requires fewer extra parameters and training costs.

**Strengths:**

- The paper proposes a unified framework for solving two challenging tasks. Although reusing the attention layers of pre-trained T2I diffusion models for various purposes has been common in recent works, such a general framework applicable to different generation tasks is still interesting.

- Extensive experimental results are presented for both tasks, and the proposed ARCI is resource-friendly, requiring only a tiny amount of extra parameters and fewer training costs than other approaches.

- The Ref-inpainting results are promising. The model is lightweight and also achieves state-of-the-art inpainting quality. The attention visualization is convincing and shows that the model learns to look at the correct locations in the reference images.

**Weaknesses:**

- The biggest issue of the paper is the writing quality, which makes the paper very hard to follow. Details are listed below.

    1. The introduction is extremely long and poorly organized. Many points are made, but I cannot find a precise sentence that emphasizes the essential contribution of the paper. The two applications (Ref-inpainting and NVS) seem to stem from the unified ARCI approach, but the introduction always tries to separate them when discussing their challenges and claiming improvements.

    2. While the introduction makes separate claims for the two tasks, the descriptions of the two tasks in Sec.3 are heavily entangled, making it hard to clearly understand how the model works for each task.

    3.  Fig.1 to Fig.3 are very difficult to parse. The texts in the figures are too small. The inputs and outputs for each task are not clearly explained. The captions are not self-contained, and it is also very hard to link them to certain parts of the main text.


- There seems to be no *quantitative evaluation* for multiview image generation -- Table 2 of the main paper only provides results with a single target view. Since the paper claims improvements in multiview image generation, it is important to formally evaluate the consistency of the generated multiview images.

- The proposed ARCI is limited by the autoregressive generation design and cannot produce many multiview images. While a potential tradeoff is to constrain the length of the condition, it would definitely sacrifice the quality/consistency of the generated images. Note that an important goal of multiview image generation is to extract the 3D object/geometry. Both the number of views and the multiview consistency are important when exporting a 3D model from the generated images. The proposed designs are suboptimal compared to recent works such as [SyncDreamer](https://arxiv.org/abs/2309.03453) and [MVDiffusion](https://arxiv.org/abs/2307.01097), which can generate 16 or more views in parallel.

- Although the authors claim that ARCI outperforms Zero123 in novel view synthesis (NVS), Zero123 itself is not merely an NVS model. Zero123 can serve as a general diffusion model backbone with 3D shape/global priors, which facilitates many other approaches via fine-tuning or score distillation (e.g., [Magic123](https://arxiv.org/abs/2306.17843), [One-2-3-45](https://arxiv.org/abs/2306.16928), [SyncDreamer](https://arxiv.org/abs/2309.03453), etc). It is true that ARCI outperforms Zero123 in NVS, especially when the training budget is limited, but the scope of ARCI is much narrower than Zero123 given the more complicated designs.

**Questions:**

As detailed in the weakness section, I believe the poor writing organization significantly impairs the quality of the paper. At the same time, the value of the proposed ARCI for multiview NVS is not convincing -- there are no quantitative evaluations for multiview image generation, and the method design seems to be suboptimal when we need to generate a large number of consistent views.

---

> ### Author Response · Authors · 2023-11-16
> **Response to Reviewer a7pP**
>
> Thanks for your valuable feedback. We recognize that the pivotal contribution of our work has not been adequately emphasized and will make a major revision to our paper to rectify this issue.
>
> **1. Entangled descriptions of the two tasks in Sec.3**
>
> Thanks for this comment. We should clarify that ARCI is a generalized model to address these two tasks.
> For the single reference-based synthesis, Ref-inpainting and NVS work very similarly in ARCI (just stitching reference and masked target). The three differences are 1) NVS needs to fine-tune the whole SD, 2) NVS needs another pose FC to encode relative pose information, 3) NVS needs to add positional encoding before the self-attention modules. We will further claim these in Sec.3.1 and Sec.3.2.
> For the multi-view synthesis, the only difference is the autoregressive generation for NVS, as detailed in Sec.3.2.
>
> **2. No quantitative evaluation for multiview image generation**
>
> Thanks. We add the multi-view quantitative evaluation as follows:
> |         | Ref view | PSNR   | SSIM  | LPIPS  | CLIP   | P-CLIP |
> |---------|----------|--------|-------|--------|--------|--------|
> | Zero123 | first    | 19.265 | 0.855 | 0.1366 | 0.7723 | 0.7756 |
> | Zero123 | last     | 14.621 | 0.767 | 0.2569 | 0.6921 | 0.7667 |
> | ours    | first    | 21.573 | 0.883 | 0.1143 | 0.7964 | 0.7709 |
> | ours    | AR       | 21.271 | 0.882 | 0.1195 | 0.7882 | 0.7958 |
>
> We introduce the pairwise CLIP score (P-CLIP) to verify the consistency of all generated samples.
> LeftRefill outperforms Zero123 in most metrics, while AR could prominently improve the consistency with just a little quality degradation.
>
> **3. ARCI is limited by the autoregressive generation design and cannot produce many multiview images**
>
> Thanks. We have provided results of consistent multi-view synthesis in our supplementary.
>
> **4. ARCI is much narrower than Zero123 given the more complicated designs**
>
> Thanks. ARCI is a simple model design that can also be used for image-to-3D prior learning. Because ARCI could generate consistent multi-view images for methods like NERF to learn. Since we do not focus on 3D generation, these works can be seen as interesting future work.

---

### Author Response · Authors · 2023-11-16
**Response to all reviewers**

We appreciate all insightful feedback from the reviewers, including positive comments like “general framework”, “promising results”, and “novel and interesting direction”. The main concern of all reviewers is the writing quality of this paper. We apologize for any confusion resulting from unclear writing, and we recognize that the pivotal contribution of our work has not been adequately emphasized. We clarify that our primary contribution is spatially stitching reference and target views together as a whole input, which can effectively improve the capability for reference-based synthesis and training convergence of large Text-to-Image models.  We will definitely make a major revision to our paper to rectify these issues. Thanks again for all your valuable comments!